# Sustainable Ketalization of Glycerol with Ethyl Levulinate Catalyzed by the Iron(III)-Based Metal-Organic Framework MIL-88A

**DOI:** 10.3390/molecules27217229

**Published:** 2022-10-25

**Authors:** Massimo Melchiorre, Domenico Lentini, Maria Elena Cucciolito, Francesco Taddeo, Maryam Hmoudah, Martino Di Serio, Francesco Ruffo, Vincenzo Russo, Roberto Esposito

**Affiliations:** 1Department of Chemical Sciences, University of Naples Federico II, Complesso Univ. M.S. Angelo, Via Cintia, IT-80126 Naples, Italy; 2CIRCC Consorzio Interuniversitario di Reattività Chimica e Catalisi, Via Celso Ulpiani 27, IT-70126 Bari, Italy; 3ISusChem S.r.l., Piazza Carità n 32, IT-80134 Napoli, Italy; 4Laboratory of Industrial Chemistry and Reaction Engineering, Åbo Akademi, Henriksgatan 2, FI-20500 Turku, Finland

**Keywords:** glycerol, ethyl levulinate, ketalization, MOF, heterogeneous catalysis

## Abstract

The catalytic properties of a simple iron-containing MOF based on fumaric acid, MIL-88A, were investigated in the ketalization of ethyl levulinate with glycerol. The corresponding product is a component of current interest as a renewable building block for many uses. Under the following conditions (solventless, 120 °C, stoichiometric ratio, 1% cat.), the reaction proceeds with good yields (85%), and the catalyst can be recovered and recycled without loss of activity, despite some changes in the crystalline lattice and morphology. Moreover, the residual iron content in the product is in the order of units of ppm (≤2), which demonstrates the robustness of the MOF under the reaction conditions.

## 1. Introduction

MOFs (metal–organic frameworks) are hybrid organic–inorganic materials where metal ions are interconnected through polydentate organic ligands. The result is a three-dimensional polymeric structure, with geometrical features that depend on the nature of both the metal ion and the organic linker. These materials are often characterized by high porosity and surface areas, thus making them a valid alternative to zeolites for several applications, such as catalysis, drug delivery, gas storage, sensors [1,2], and pollutant adsorbents [3].

Their use as heterogeneous catalysts has been recently claimed in several reports. An accurate design allows for mimicking the active coordination environments of metal ions, and, therefore, promoting catalysis in the same way as the corresponding homogeneous complexes. At the same time, the insoluble nature of MOFs allows for their immediate separation and reuse, as in the case of classic heterogeneous catalysts. The latter is a key aspect in determining the sustainability of an industrial process because the difficulty of recovering the catalyst is often the limiting step in its development, which justifies the distrust that industry shows toward homogeneous catalysis.

At the same time, the effectiveness of this approach requires robustness of the MOFs in the reaction conditions, to prevent leaching of the metal ions, which would imply homogeneous contributions to catalysis and undesired contamination of the product. This possibility is not marginal, if we consider that studies on MOFs have already highlighted their tendency to degrade and lose their original structure when stressed. However, this aspect does not always seem to be addressed with due attention, and, therefore, the state of the art does not yet allow the fully validated use of MOFs in catalysis.

Intending to contribute to the understanding of this delicate theme, we have undertaken a study on the catalytic properties of MOF MIL-88A in the ketalization of ethyl levulinate with glycerol to produce **1** (Figure 1), a product that is currently arousing growing interest as a renewable building block for many uses [4].

Ethyl levulinate was chosen, instead of methyl levulinate, due to the more sustainable alcohol involved during its synthesis (ethanol instead of methanol).

This process also fully matches other sustainability requirements, which implies [5]: (1) maximizing the *atom economy* of a reaction, via reaction paths with no byproducts, additives, or solvents [6,7]; (2) reintroducing a byproduct as its raw material, i.e., glycerol, the byproduct of the synthesis of biodiesel [8].

The presence of three vicinal hydroxyl groups allows for exploitation in the synthesis of various molecules such as diols ethylene glycol, acrolein, glycerides, acetals, ketals, and syngas through the related catalyzed reactions [9,10].

Levulinic acid, MeC(O)CH_2_CH_2_CO_2_H, and its esters, MeC(O)CH_2_CH_2_CO_2_R, also represent important renewable sources of synthons due to their accessibility from lignocellulosic biomass. Similarly, other esters derived from biomass [11,12] are currently used in industry as precursors for solvents, fuel additives, cosmetics, and food additives [13,14,15,16,17,18,19]. In addition to esterification, the formation of the corresponding ketals also represents a way to obtain molecules with added value. Their versatility, also due to the intermediate polarity between the starting ketone and glycerol, allows them to have several applications such as gasoline additives [20,21,22] and in detergent formulation [23].

Since the ketalization of Figure 1 is an equilibrium reaction, the removal of water is a crucial step to ensure ketal formation. This can be achieved using molecular sieves or distillation or by operating at temperatures higher than 100 °C. Furthermore, an acid catalyst is required to fasten the reaction. Traditionally, the reaction has been carried out with Brønsted acid catalysts, but their well-known corrosive action has persuaded the scientific community to search for more suitable systems. Therefore, Lewis acid catalysts based on transition metal ions have acquired increasing interest, given the greater tolerance of the reaction system. However, their choice must also be evaluated, as the use of precious metals and/or the too-high loading are not convenient and make separation difficult. Cheap, non-toxic, abundant, and well-distributed metals should be chosen. Some of these have already been successfully used for the conversion of biomass and derivatives into useful products by oxidation [24,25], esterification [26,27], transesterification [26,28], and other reactions. Among them, iron stands out for its cost, high availability, and low toxicity. Some of its compounds and simple salts are also very active in ketalization reactions [29,30,31], and many ketals were prepared using iron(III)-based homogeneous catalysts, including a methyl levulinate ketal.

While there are examples of the catalyzed ketalization of glycerol with methyl levulinate in the literature [32,33,34], there are fewer examples with ethyl levulinate, most of which are patents [4,35,36]. Table 1 summarizes the results found in the literature.

In none of the cases shown in Table 1, where a heterogeneous catalyst is used, was recycling of the catalyst attempted. The paper from Mullen et al. [4] is the one case in the non-patent literature that reports the synthesis of **1** and, in the same paper, the authors also report the use of Amberlyst 15 as a heterogeneous catalyst, though the catalyst loading is not reported and, again, no recycles of the catalyst was attempted.

Following our results on the activity of iron(III)-based homogeneous catalysts, in this paper, the ketalization of glycerol with ethyl levulinate has been carried out using the simplest MOF containing iron(III), MIL-88A. The study aimed to verify the performance of the catalyst according to temperature, catalyst loading, and ratio between the reactants. Moreover, the morphological aspect of the spent catalyst has been evaluated. The results show that in the best conditions the catalysis proceeds conveniently, and that the MOF can be recovered and reused effectively for five successive runs.

The quantity of Fe(III) ions found in the product is always less than a few ppm, a result that demonstrates the heterogeneous contributions of the catalyst, even if the MOF shows morphological changes during their recycles in both the highest and lowest reaction temperatures used (120 °C and 75 °C).

## 2. Results and Discussion

### 2.1. Catalyst Synthesis and Characterization

The catalyst was prepared as reported by Chelati et al. [37] The preparation is simple, involves mixing equimolar quantities of iron(III) chloride hexahydrate and fumaric acid (Figure 2), and requires only water as the solvent.

The XRD pattern in Figure 1 shows strong peaks at 2θ positions 10.5° and 11.9°, confirming the crystallinity of the prepared MIL-88A. SEM images (Figure 1, right) demonstrate the uniform rod-like-shaped crystals with a length between 1.3 and 2.6 μm and a width of about 400 nm, in accordance with the literature [38,39]. Thermogravimetric analysis (TGA, Figure 2) was also used, and it is possible to distinguish three main peaks: the first at 71.7 °C, probably due to the residues of acetone used in the washing; the second at 110 °C, due to the desorption of water; and the third at 183 °C, which suggests the degradation of the MOF. This attests to the thermal stability under the applied conditions.

In addition, the BET surface area of 9.9 ± 0.2 m^2^/g (Figure 3) is in line with the result obtained in the literature for MIL-88A under similar preparations [40,41]. The material is microporous, as expected, and the BET diagram shows an isotherm resembling type IV with a little pronounced hysteresis of type H3. The synthesized catalyst pores diameter < 2 nm (in the structure the diameter is about 10 Å [42]), as shown by BJH analysis (Appendix A), where no pores are present in the range from 5 to 50 nm.

### 2.2. Catalyst Optimization and Recycling

The evaluation of the performances of MIL-88A as a catalyst in the ketalization of ethyl levulinate with glycerol was conducted by screening the operating temperature, the molar ratio ethyl levulinate/glycerol, and the catalyst loading (expressed as % mol catalyst/mol of glycerol). In every experiment set, the recyclability of the catalyst was also evaluated. After the end of each reaction (the first made with a fresh catalyst was named Run 0), the catalyst was recovered by centrifugation, washed, and reused in subsequent runs (from Run 1 to a maximum of Run 5).

The effect of the temperature was evaluated by screening the reaction at 120 °C, to favor the removal of water from the reaction mixture, and at 75 °C and 100 °C as a milder condition. The first tests were carried out in test tubes (open system), with a stoichiometric ratio of ethyl levulinate and glycerol (1:1), in the absence of solvents, at the desired temperature, with a 1% mol of catalyst loading. In these conditions, the yields obtained after 22 h were: 37% at 75 °C, 63% at 100 °C, and 84% at 120 °C. These results show that working at a temperature lower than or equal to the boiling point of water combines the drawback of reducing the reaction rate with inhibiting the formation of products due to a less effective removal of the water.

Subsequent investigations were carried out at 120 °C. The effect of the catalyst loading was evaluated by performing a test at 0.1% mol, carried out in test tubes (open system), with a stoichiometric ratio of ethyl levulinate and glycerol (1:1), in the absence of solvents at a temperature of 120 °C.

The reaction reached 78 ± 3% yield after 22 h (recycled runs shown in Appendix A). In the same conditions, the blank test produced a yield of 64 ± 3%. The reaction selectively favors (>98%) the dioxolane five-member ring **1**, even without the catalyst (Appendix A shows the NMR spectrum of a reaction at low conversion, where the signals of six-member ring **2** are detectable). This distribution is in agreement with the literature findings [43] on glycerol ketalization, which demonstrate how the selectivity toward type **1** species increases with temperature. Plausibly, the six-membered ring is the kinetic product of the reaction, and it is, therefore, present only at low temperatures or in the initial stages of the reaction.

The catalyst loading was then increased to 1%, in moles, leaving all the other conditions unchanged. Samples were analyzed after 1, 6, and 22 h of reaction, obtaining the yields reported in Table 2.

After 22 h of reaction, a yield of 84 ± 3% remains constant upon recycling, suggesting its reusability under these reaction conditions.

In a subsequent set of experiments, the catalyst concentration was brought up to 5%, while all the other reaction conditions were kept constant (T = 120 °C; mol ethyl levulinate: mol glycerol = 1:1).

After 6 h (Table 3), there is no beneficial effect attributable to the increase in catalyst loading. A slight decrease in yields is even observed, which is probably attributable to the increase in the viscosity of the reaction mixture.

A further set of experiments was conducted by increasing the molar ratio between ethyl levulinate and glycerol to 4:1.

The yield after 6 h is slightly higher than that observed with the equimolar ratio (Table 4). Most likely, the potential beneficial effect of the excess of one of the reactants is offset by the dilution of the catalyst. These results highlight that the use of a non-stoichiometric ratio is not convenient for both the reaction rate and atom economy aspects. Therefore, further studies about different molar ratios were not undertaken.

In the best reaction conditions (120 °C; reactants molar ratio 1:1, catalyst loading of 1%, in moles, with respect to glycerol), the catalyst was used in consecutive runs and recycled up to five times, showing the same kinetic profiles in each experiment (Figure 4 and Appendix A, Runs 0–5).

Leaching of the metal ion was evaluated by UV–Vis titrations and resulted in less than 2 ppm in each recycle (Appendix A). This is a crucial analysis to estimate both the stability of the catalyst in the long run and the homogeneous catalytic contribution due to leaching phenomena. To evaluate the homogeneous catalytic contribution of leached Fe(III) ions, a catalytic test was performed using FeCl_3_ in a leaching-comparable concentration (2 ppm). The kinetic profiles are shown below (Figure 4, Appendix A). They demonstrate that, in this case, the catalytic activity of the Fe(III) ion in homogeneous conditions is negligible.

To further investigate the stability of the catalyst, SEM images of the spent MOF (Appendix A, Run 2) were acquired. Round-edged particles (Figure 5B) were found in contrast with the rod-shaped fresh sample. The diffraction pattern recorded within the recycles also showed the loss of crystallinity of the catalyst (Figure 5A). However, the catalytic activity seems to be unaffected by these changes, as it remains constant during the successive recycles. Several papers were published on the topic, demonstrating that even if during the reaction a structure change of the catalyst can be observed, e.g., a change in the XRD pattern or morphology, the activity remains the same. For example, Cheng et al. demonstrated that CO_2_ adsorption is not affected by the structural changes that a Cu-MOF undergoes during water exposure [44]. The reason behind this phenomenon is still under investigation and is strictly related to the specific chemical application.

To verify the effect of milder conditions upon the crystallinity of the catalyst, a screening was performed at 75 °C, keeping the other conditions unaltered (reactants’ molar ratio equal to 1:1, MIL-88A loading of 1% mol). Moreover, also in this case, a comparative run using a homogeneous catalyst (FeCl_3_, loading 2 ppm) was performed (Figure 6).

Also at this temperature, the MIL-88A is more active than the Fe(III) ion in homogeneous conditions, and the catalytic activity is stable despite the loss of crystallinity (Figure 7).

From these results, MIL-88A is an active catalyst in the ketalization of glycerol with ethyl levulinate, and good yields can be reached in 22 h at a moderate temperature. The contribution of homogeneous iron(III) is negligible in Run 0, where small residues of the iron(III) salts used as the reactant are probably washed away. The following runs do not show appreciable leaching, instead showing reproducible and stable activity.

These results are of particular interest if one bears in mind that no water-removal technology has been used, unlike what is reported in the literature (Table 1), and no solvent or auxiliary substances are used either.

## 3. Materials and Method

### 3.1. Materials

All the reagents used in the experiments and catalyst preparation reagents were purchased from Merck at the highest purity level.

### 3.2. Catalyst Preparation

**MIL-88A.** The catalyst MIL-88A was prepared, as indicated by Chalati et al. [37], following what they refer to as an ambient pressure dynamic hydrothermal synthesis; in a 500 mL round bottom flask iron(III) chloride hexahydrate (13.5 g, 50.0 mmol), fumarate acid (5.80 g, 50.0 mmol) and 250 mL of water were added. The flask was equipped with an Allihn condenser, and the mixture was bought to reflux and kept at that temperature for 30 min under continuous stirring. The flask was allowed to cool at room temperature, and then the mixture was centrifuged at 3500 rpm for 20 min and washed two times with fresh water, two times with ethanol, and once with acetone. Every time, the solid was recovered by centrifugation, to eventually isolate a pale orange powder (6.45 g, yield: 63%).

X-ray diffraction pattern (XRD) was used to investigate the crystallinity of the dried powder of MIL-88A. X-ray PANalytical Diffractometer (Malvern, Worcestershire, UK) diffraction system was used, equipped with a nickel filter Cu Kα radiation operating at 40 kV and 40 mA with a *θ*–2*θ* goniometer. The MIL-88A powder was placed on a zero-background holder, and the scan range was 5–60° 2*θ* degrees using a 0.05° step and a counting time of 187 s per step. Morphological properties of MIL-88A were investigated using scanning electron microscopy (SEM) technique. SEM images were observed by FEI Nova NanoSEM 450 at an accelerating voltage of 5 kV with Everhart Thornley Detector (ETD) and Lens Detector (TLD) at high magnification. Thermogravimetric analysis (TGA) was performed in presence of airflow of 100 cm^3^/min and a heating rate of 10 °C/min using TG/DTG analyzer (STGA-1000, SANAF, Istanbul, Turkey). The TGA sample mass was kept low (~5 mg) to avoid diffusion limitations.

Surface and micropore analysis was carried out with a Micrometrics ASAP 2000.

### 3.3. Catalysts Activity and Reuse Tests

An appropriate amount of the catalyst, 0.920 g of glycerol (10.0 mmol), and an appropriate amount of ethyl levulinate (1.44 g, 10.0 mmol or 5.76 g, 40.0 mmol) were added in a 10 mL open test tube equipped with a magnetic stirrer. The reaction temperature was set and kept constant, monitored through a thermometer immersed in a preheated heating bath. The tube was immersed in the bath, and this time was considered the starting point of the reaction; from this moment, samples were withdrawn at regular intervals to be analyzed through ^1^H NMR.

After the last sample was withdrawn, the tube was allowed to cool at room temperature, and then it was centrifuged to remove the reaction mixture as supernatant and to quasi-quantitively (94–99% wt) recover the catalyst by avoiding solid transfers. Then, it was washed in the same tube one time with fresh ethanol and two times with acetone. Every time, the catalyst was recovered by centrifugation at 3500 rpm for 20 min. The solid was allowed to dry in an oven at 60 °C overnight, and then it was weighed and reused in a successive recycle run with the appropriate amount of fresh glycerol and ethyl levulinate in the same tube, following the procedure described previously.

### 3.4. Yield Calculation

Samples were analyzed through ^1^H NMR by a Bruker Avance Ultrashield operating at a proton frequency of 400 MHz using CDCl_3_ or D_2_O as solvent.

The conversion was obtained by integrating the signals of methyl groups (two, because there are two diastereoisomers) resonating at a chemical shift between 1.37 and 1.33 ppm (in blue in Figure 8) and comparing this integral (*I_P_*) with the integral (*I_M_*) of the signal resonating between 1.30 and 1.19 ppm (in red in Figure 8) that corresponds to the signal of the protons of terminal methyls of the ethyl group (in red in Figure 8) of both the reactant (ethyl levulinate) and product (**1**). The molar fraction of the product (*X_Ket_*) was calculated as follows:(1)$$X_{Ket}=\frac{I_{P}}{I_{M}}$$

A confirmation of the calculated yield can be obtained by also evaluating the integrals of the two triplets of the -C*H*_2_C*H*_2_- portion of ethyl levulinate (in green in Figure 8).

**Characterization: 1** (ethyl 3-(4-(hydroxymethyl)-2-methyl-1,3-dioxolan-2-yl)propanoate). ^1^H NMR (500 MHz, CDCl_3_, two diastereoisomers) δ 4.27–4.17 (m, 1H), 4.13 (q, *J* = 7.2 Hz, 2H), 4.08–3.95 (m, 1H), 3.89–3.71 (m, 2H), 3.61–3.50 (m, 1H), 2.50–2.31 (m, 2H), 2.09 (t, *J* = 6.9 Hz, 1H), 2.01 (t, *J* = 7.6 Hz, 1H), 1.37 and 1.33 (s, 3H), 1.30–1.19 (m, 3H).

### 3.5. Iron Content Measurement

Free iron(III) content in the samples was evaluated through UV–Vis spectroscopy. The metal was analyzed by titrating it as iron thiocyanate complex [Fe(SCN)_6_]^3−^ and quantified by the method of standard additions. At the end of each catalytic run, the catalyst was removed by centrifugation, and the supernatant corresponding to the reaction crude was filtered on celite. An aliquot of 1.00 mL of this mixture was withdrawn, and the sample was brought to a volume of 2.00 mL by adding a solution of 0.20 M of thiocyanate in a 1:1 mixture of acetone and water, where the latter was buffered at a pH of 2.5, obtaining **Sample A**. A **blank** was prepared by bringing 1.00 mL of the filtered reaction crude to a volume of 2.00 mL by adding a 1:1 mixture of acetone and water, where the latter is buffered at a pH of 2.5. A UV–Vis spectrum of **Sample A** was acquired vs. the **blank**, and the absorbance at 484 nm was registered. Then, 5 µL of a solution 17.7 mM of Fe(III) was added to **Sample A** and another UV–Vis spectrum vs. **blank** was acquired, registering the absorbance at 484 nm. The procedure was repeated another two times, affording four-points calibration curve that was linearly fitted. The R^2^ resulted higher than 0.995 in all the experiments. The initial iron(III) concentration was calculated by linear regression.

## 4. Conclusions

In this work, the performance of the catalyst belonging to the family of MOFs (metal–organic frameworks) based on Fe(III), MIL-88A, in the ketalization reaction between ethyl levulinate and glycerol was investigated. The reaction was carried out in the presence of stoichiometric quantities of the reagents, in solvent-free conditions, and without the use of a vacuum or controlled atmosphere. In the best conditions, the results were quite satisfactory and very competitive with those present in the literature, as yields greater than 80% were recorded and are also preserved in subsequent reuses of the catalyst, even if a strong structural change of the catalyst was observed.

In general, the reaction studied provides that: (1) the reagents are obtainable from renewable sources, such as biomass or byproducts of its processing; (2) a heterogeneous and recyclable catalyst based on Fe(III) is used, which has a favorable ecotoxicological profile; (3) the mechanism of action of MOFs, in this case, is essentially heterogeneous, and a negligible physiological leaching of the catalyst has a minor effect on catalysis; (4) the product is an additive widely used in different types of industry. These aspects pave the way to sustainable industrial applications that maximize the atom economy and do not require further purification steps.

This study proves the feasibility of using MOFs in the heterogeneous catalysis of the ketalization of glycerol. It opens the door for further studies aimed to exploit MOFs’ 3D tunable structure to obtain a finer control of regio- and stereoselectivity.

## Data Availability

The data presented in this study are available in Appendix A.

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
