# Peer review of "Sustainable Ketalization of Glycerol with Ethyl Levulinate Catalyzed by the Iron(III)-Based Metal-Organic Framework MIL-88A"

_molecules, 2022, doi:10.3390/molecules27217229_

Round 1
Reviewer 1 Report
This paper is a typical one in Catalysis, where a catalyst is prepared and characterized, and then used in a model reaction. In this case, the reaction is the ketalization of ethyl levulinate with glycerol.
As far as the catalyst characterization, I miss a more complete morphological one. The authors share the surface area, but something else should be mentioned, i.e., type of isotherm. Particularly, pore size is very important to elucidate if pore surface is accessible to reactants or not. On the other hand, particle size should be quoted explicitly in the manuscript
Regarding the performance experiments, the authors claim that some experiments were carried out at optimized conditions. However, in this paper an optimization study is not performed. In any case, the authors can say that the best results were found at the following conditions …
The effect of temperature, and initial reactor composition was only tested at two temperatures and two initial reaction compositions. In my opinion a third temperature and third initial reaction composition are necessary to say that the influence of the two parameters is studied.
From Tables 2 and 3, and Figure 4 it is seen that the effect of the homogeneous reaction is noticeable in the reaction conditions, also that iron is quite effective as catalyst without MOF structure. Therefore, which is the real applicability of the MIL-88A catalysts?
Finally, a comparison of obtained results with those in the open literature are welcome
Author Response
We would like to thank the reviewer for their useful critiques to enhance the presented work. Below is a detailed response to all suggestions and comments.
Reviewer #1: This paper is a typical one in Catalysis, where a catalyst is prepared and characterized, and then used in a model reaction. In this case, the reaction is the ketalization of ethyl levulinate with glycerol.
- As far as the catalyst characterization, I miss a more complete morphological one. The authors share the surface area, but something else should be mentioned, i.e., type of isotherm. Particularly, pore size is very important to elucidate if pore surface is accessible to reactants or not. On the other hand, particle size should be quoted explicitly in the manuscript
Answer: The MIL88A appears as crystalline material with micropores, as expected. The diameter and volume of pores of such materials are usually outside the range of investigation of the BET, about this topic reference 42 is also now added: “Serre, C.; Mellot-Draznieks, C.; Surblé, S.; Audebrand, N.; Filinchuk, Y.; Férey, G. Role of Solvent-Host Interactions That Lead to Very Large Swelling of Hybrid Frameworks. Science 2007, 315, 1828-1831, doi:10.1126/science.1137975”. For this reason, BET description has been extended and the BJH analysis has been reported and added as Figure S2. In addition, the size of crystallites is added on line 122 (page 5).
- Regarding the performance experiments, the authors claim that some experiments were carried out at optimized conditions. However, in this paper an optimization study is not performed. In any case, the authors can say that the best results were found at the following conditions …
Answer: As suggested by the referee, the term “optimized” was changed in the text.
- The effect of temperature, and initial reactor composition was only tested at two temperatures and two initial reaction compositions. In my opinion a third temperature and third initial reaction composition are necessary to say that the influence of the two parameters is studied.
Answer: An experiment was also performed at the intermediate temperature of 100 °C, and the results are now reported at page 6 with some comments: “The effect of the temperature was evaluated by screening the reaction at 120 °C, to favor the removal of water from the reaction mixture, and at 75 °C and 100 °C as milder condition. The first tests were carried out in test tubes (open system), with a stoichiometric ratio of ethyl levulinate and glycerol (1:1), in the absence of solvents, at the desired temperature, with a 1 % mol of catalyst loading. In these conditions the yields obtained after 22 h were: 37% at 75 °C, 63% at 100 °C and 84% at 120 °C. These results show that working at a temperature lower than or equal to the boiling point of water combine the drawback of reducing the reaction rate and inhibiting the formation of products due to a less effective removal of the water.
Subsequent investigations were carried out at 120°C. The effect of the catalyst loading was evaluated performing a test at 0.1 % mol, carried out in test tubes (open system), with a stoichiometric ratio of ethyl levulinate and glycerol (1:1), in the absence of solvents at a temperature of 120 °C.”
A comment about the molar ratio investigation is also added at page 7: “These results highlight that the use of non- stoichiometric ratio is not convenient for both, reaction rate and atom economy aspects. Therefore, further studies about different molar ratios were not deepened.”
- From Tables 2 and 3, and Figure 4 it is seen that the effect of the homogeneous reaction is noticeable in the reaction conditions, also that iron is quite effective as catalyst without MOF structure. Therefore, which is the real applicability of the MIL-88A catalysts?
Answer: This is a very relevant point, central to this study that aimed to evaluate the real applicability of this catalytic system. In fact, we wanted to investigate the catalytic process to evaluate both the expected heterogeneous contribution and a possible homogeneous one due to possible leaching of iron ions. This is rarely addressed in the literature, but it is important for the correct interpretation of studies in this emerging sector of catalysis. Our results, as illustrated within the text, demonstrate that the mechanism of action of MOFs is essentially heterogeneous, and that a negligible physiological leaching of the catalyst has a minor effect on catalysis.
These considerations were added in the conclusion (page 13).
- Finally, a comparison of obtained results with those in the open literature are welcome.
Answer: As suggested by the referee, some literature results were already collected in Table 1.
Reviewer 2 Report
The authors synthesized the MOF (MIL88A) and characterized the MOF using XRD to confirm the crystallinity, TGA for thermal stability, SEM images to find out morphologies and BET analysis to find out the porosity and surface area. They used MOF as a catalyst in ketalization of ethyl levulinate with glycerol. The catalytic performance was studied by measuring the yield of ketal at diffident time intervals of reaction and at 120 and 75 °C. They also reported a yield of 84% remains constant for above five times recycling of the catalyst. They also studied the effects of the ratio of ethyl levulinate vs. glycerol and different catalyst loading. The authors also pointed out that their approach can be used in industrial applications without the need of solvent and vacuum conditions. I think the paper can be accepted for publication with minor revision.
However, the authors should address the following comments.
1. The authors stated in line 75 “The removal of water is a crucial step to ensure ketal formation,” and in line 240 “These results are of particular interest if one bears in mind that no water removal technology has been used.” Please explain why water removal is not necessary in this work?
2. The authors stated that recycling reduces crystallinity (see figure 5). If so, how the recyclability of catalyst is not affecting the yield?
3. The authors mentioned that increase in catalyst loading causes a decrease in yields due to an increase in the viscosity of the reaction mixture. Please provide the viscosity of ethyl levulinate and glycerol at 120 °C, if available.
Author Response
We would like to thank the reviewer for their useful critiques to enhance the presented work. Below is a detailed response to all suggestions and comments.
Reviewer #2: The authors synthesized the MOF (MIL88A) and characterized the MOF using XRD to confirm the crystallinity, TGA for thermal stability, SEM images to find out morphologies and BET analysis to find out the porosity and surface area. They used MOF as a catalyst in ketalization of ethyl levulinate with glycerol. The catalytic performance was studied by measuring the yield of ketal at diffident time intervals of reaction and at 120 and 75 °C. They also reported a yield of 84% remains constant for above five times recycling of the catalyst. They also studied the effects of the ratio of ethyl levulinate vs. glycerol and different catalyst loading. The authors also pointed out that their approach can be used in industrial applications without the need of solvent and vacuum conditions. I think the paper can be accepted for publication with minor revision.
However, the authors should address the following comments.
- The authors stated in line 75 “The removal of water is a crucial step to ensure ketal formation,” and in line 240 “These results are of particular interest if one bears in mind that no water removal technology has been used.” Please explain why water removal is not necessary in this work?
Answer: There is no need for measures dedicated to the removal of water, because the system is open, and the removal of the water is due to the reaction temperature (see page 6). This circumstance is now specified also in the experimental section.
- The authors stated that recycling reduces crystallinity (see figure 5). If so, how the recyclability of catalyst is not affecting the yield?
Answer: This aspect is certainly noteworthy, and has already been discussed in the original manuscript with some comments reported in lines 219-229
- The authors mentioned that increase in catalyst loading causes a decrease in yields due to an increase in the viscosity of the reaction mixture. Please provide the viscosity of ethyl levulinate and glycerol at 120 °C, if available.
Answer: The increased viscosity of the system is due to the very high amount of solid added to the mixture. Reaching the 5% in mol with respect to glycerol a pasty material is obtained. So, the cause of increased viscosity is the presence of high amounts of solid catalysts. At that loading, the magnet is barely able to stir the mixture.
Round 2
Reviewer 1 Report
The authors have answered most of my queries. The paper now contains more information, and this information is enough to justify its publication. Next, you can find a short list of minor points to be addressed before publication.
1) Page 3, lines 109-110. It says “… even if the MOF shows morphological changes during their recycles in both of the reaction temperatures used (120°C and 75°C)”. This sentence is not clear since the authors have performed experiments at 100°C too.
2) Page 3, line 116 and page 11, line 264. The authors speak of fumarate acid, but structure of scheme 2 correspond to fumaric acid.
3) In the caption of Figure 7, it is quoted that the catalyst is used at 75°C. However, in the figure appears a temperature of 70°C. It seems to me that temperature of 70°C is not correct.
4) Page 12, Figure 8 and lines 304-306. In the text, it is said that chemical shift between 1.37 and 1.33 ppm are in red in Figure 8, and signals between 1.3 and 1.19 appear in blue in figure 8. However, in Figure 8 signals between 1.37 and 1.33 are in blue, and signals between 1.3 and 1.19 are in red. Please correct
Author Response
We thank the referee for finding these typos in the manuscript.
Here there is the response point by point.
1) Page 3, lines 109-110. It says “… even if the MOF shows morphological changes during their recycles in both of the reaction temperatures used (120°C and 75°C)”. This sentence is not clear since the authors have performed experiments at 100°C too.
Answer: we reformulated into: "...during their recycles in both lowest and highest reaction temperatures used (120 °C and 75 °C)."
2) Page 3, line 116 and page 11, line 264. The authors speak of fumarate acid, but structure of scheme 2 correspond to fumaric acid.
Answer: we corrected the typo.
3) In the caption of Figure 7, it is quoted that the catalyst is used at 75°C. However, in the figure appears a temperature of 70°C. It seems to me that temperature of 70°C is not correct.
Answer: we corrected the typo.
4) Page 12, Figure 8 and lines 304-306. In the text, it is said that chemical shift between 1.37 and 1.33 ppm are in red in Figure 8, and signals between 1.3 and 1.19 appear in blue in figure 8. However, in Figure 8 signals between 1.37 and 1.33 are in blue, and signals between 1.3 and 1.19 are in red. Please correct
Answer: we corrected the typo.